# How large should a cause of death be in order to be included in mortality trend analysis? Deriving a cut-off point from retrospective trend analyses in 21 European countries

Marianna Mitratza ,[1] Jan W P F Kardaun,[1,2] Anton E Kunst[1]

[1]Department of Public Health, Amsterdam UMC, University of Amsterdam, Amsterdam, The Netherlands
[2]Department of Health and Care, Statistics Netherlands, The Hague, The Netherlands

**Correspondence to**
Dr Marianna Mitratza;
m.mitratza@amsterdamumc.nl

## ABSTRACT

**Objectives** The International Classification of Diseases (ICD-10) distinguishes a large number of causes of death (CODs) that could each be studied individually when monitoring time-trends. We aimed to develop recommendations for using the size of CODs as a criterion for their inclusion in long-term trend analysis.

**Design** Retrospective trend analysis.

**Setting** 21 European countries of the WHO Mortality Database.

**Participants** Deaths from CODs (3-position ICD-10 codes) with ≥5 average annual deaths in a 15-year period between 2000 and 2016.

**Primary and secondary outcome measures** Fitting polynomial regression models, we examined for each COD in each country whether or not changes over time were statistically significant (with $\alpha=0.05$) and we assessed correlates of this outcome. Applying receiver operating characteristicROC curve diagnostics, we derived COD size thresholds for selecting CODs for trends analysis.

**Results** Across all countries, 64.0% of CODs had significant long-term trends. The odds of having a significant trend increased by 18% for every 10% increase of COD size. The independent effect of country was negligible. As compared to circulatory system diseases, the probability of a significant trend was lower for neoplasms and digestive system diseases, and higher for infectious diseases, mental diseases and signs-and-symptoms. We derived a general threshold of around 30 (range: 28–33) annual deaths for inclusion of a COD in trend analysis. The relevant threshold for neoplasms was around 65 (range: 61–70) and for infectious diseases was 20 (range: 19–20).

**Conclusions** The likelihood that long-term trends are detected with statistical significance is strongly related to COD size and varies between ICD-10 chapters, but has no independent relation to country. We recommend a general size criterion of 30 annual deaths to select CODs for long-term mortality-trends analysis in European countries.

## Strengths and limitations of this study

► The first study to develop a criterion to select causes of death for monitoring purposes based on their annual number of deaths.
► The analysis of a large sample of causes of death covering most European countries, using the WHO Mortality Database.
► Criteria for selection of causes were derived for different types of causes of death.
► Other criteria were not applied, such as causes of death that involve high healthcare costs or that are potentially modifiable.

for these data are the statistics maintained by national statistical offices.[1 2] National statistics of causes of death (CODs) include many codes of the 10th revision of the International Classification of Diseases (ICD-10 codes).[3] Given the detail of this classification—there are 1752 three-position ICD-10 codes—a part of it may not be instrumental for monitoring long-term time-trends due to the small number of deaths for specific codes.

When using these statistics to monitor long-term trends in mortality, a main question is which of the many possible CODs to include. At the very least, the selection should include only CODs that are large enough to have a reasonable probability of detecting a long-term mortality trend. This probability may be influenced by several factors. One main factor is the COD size, defined as the mean annual number of deaths, which expresses the rarity of a disease or condition that is selected as underlying COD in a population. Incidence changes or effects of interventions are common factors discussed in mortality trends analyses.[4 5] In addition, this probability might depend on other factors, such as the type of COD, or the country of interest. Certain types

## INTRODUCTION

Mortality data are essential for the monitoring of population-wide trends in a large number of diseases and injuries, as well as for the evaluation of health policies. A common source

of CODs may be more likely to present a long-term trend. For example, neoplasms have been shown to be more gradual in their annual changes,[6] whereas infectious diseases[7] may have high year-to-year variation. As regards to different populations, the likelihood to detect a long-term trend for a COD may vary between countries because of differences in population size, COD coding practices that may also influence observed mortality trends,[8] trends in prevalence of risk factors,[9–12] implementation of new prevention strategies,[13 14] treatment protocols[5] or health-care reforms.[15]

Due to the fact that the likelihood to detect a long-term trend of a COD may depend on various factors, there is a need for an empirical assessment of such likelihood. Such analysis may provide an empirical basis for the identification of CODs for which long-trends are likely to be detectable. More specifically, it may be used to define a criterion, or rule of thumb, that identifies eligible CODs in terms of a minimum COD size. When such a criterion allows for variation by COD type and country, it may be used in national and international trend analysis across a broad range of CODs.

The general objective of this study was to determine a COD size criterion for the study of long-term mortality trends in European countries. The specific objectives were: (1) to assess the association between the size and the type of a COD and the probability of detecting a long-term trend in European countries, (2) to assess how this association varies according to country and (3) to identify a minimum annual number of deaths recommended to monitor trends in cause-specific mortality.

## METHODS
### Data
We used annual mortality data for 21 European countries of the WHO Mortality Database (1 October 2017 update).[16] We included the 21 countries of the European Union (28 countries) or the European Free Trade Association (4 countries) that had been using ICD-10 (3 or 4 position) coding for at least 15 consecutive years. Iceland, Luxemburg and Malta were excluded because of their small population.[17] The most recent 15 year period was selected, which was 2001–2015 for all countries with few exceptions (Belgium, France and Switzerland: 2000–2014; Austria: 2002–2016). If the time series of a COD in a country was interrupted by a year without any data on that COD, we assumed that zero cases occurred.

### Statistical analysis
For each year and COD in a country, we calculated an age-standardised count of deaths using the direct method. As reference population, we used the age-distribution of the European Standard Population 2013, scaled to the mid-period population of each country. This method intended to compensate for annual changes in the age-distribution of the population, while keeping the age-standardised count close to the observed absolute numbers.

For further analysis, we analysed CODs that had at least five average age-standardised annual deaths, because most of the smaller CODs had predominantly zero or only zero annual deaths.

Long-term time-trends of the age-standardised count of deaths of each COD in each country were analysed using ordinary least squares regression (OLS) models. Trends were fitted by applying linear regression models with polynomial terms of year as continuous, independent covariates.[18] We used orthogonal polynomials in order to account for multicollinearity of the polynomial components.[19] We fitted four models: the constant, the linear, the quadratic and the cubic model (with zero, first, second and third degree polynomials, respectively). The four models were applied for all CODs in each country. We used the lowest corrected Akaike Information Criterion to select the best model for each COD in each country.[20] In a next step, the best model was compared with the constant model using the F-test, at the significance level of $\alpha=0.05$. If the best model performed better than the constant model with statistical significance, it was kept as the final best model. Otherwise, the constant model was selected as the best model for this COD. In the rest of the paper, the constant model is referred to as the absence of a demonstrable trend.

Next, using a multilevel logistic regression model, we determined how the categorisation of a COD as having a statistically significant trend (ie, best model being the linear, quadratic or cubic model) was related to COD size and COD type. These variables were included in the model as fixed effects. The COD size was defined as the mean annual number of deaths and the COD type was defined as the ICD-10 chapter in which it is classified. The chapter of circulatory diseases was the reference category, as it had the largest number of deaths. As the distribution of the number of deaths across CODs was highly skewed, we used its natural logarithm as a measure of COD size. The model also included the level of countries as random effect, in order to investigate the variation of European countries in the likelihood of detecting a long-term trend. We calculated the Intraclass Correlation Coefficient (ICC), which expresses the proportion of the variance in the outcome that is attributable to variations between the countries.[21] The ICC was calculated both with and without controlling for the fixed effects of the size and type of the COD.

Finally, we used receiver operating characteristic (ROC) curve diagnostics[22–24] to derive COD size thresholds for detecting a long-term time-trend. We calculated the Area Under the Curve (AUC) of the logistic model with COD size as the predictor and the binary categorisation of a COD as having a significant long-term time-trend as the outcome. We derived the COD size thresholds using three indices. First, we used the maximum Youden index[25–27] which represents the point of the ROC curve with the maximum sum of sensitivity (se) and specificity (sp). Second, we used the index measuring the minimum difference between sensitivity and specificity.[23] Third, we

**Table 1** Frequencies of causes of deaths according to estimates of their long-term time-trend in 21 European countries

| Country (sorted by mean population) | CODs* analysed (n) | Type of long-term time-trend (%) | | | | Mean population (thousands) | Mean crude annual deaths from all CODs together | Mean crude annual deaths from CODs analysed* |
|---|---|---|---|---|---|---|---|---|
| | | No significant trend | Linear | Quadratic | Cubic | | | |
| Estonia | 202 | 34.7 | 35.6 | 15.3 | 14.4 | 1343 | 16642 | 16006 |
| Slovenia | 228 | 43.9 | 29.4 | 17.1 | 9.6 | 2029 | 18830 | 18163 |
| Latvia | 259 | 38.2 | 28.6 | 17.0 | 16.2 | 2157 | 30840 | 30091 |
| Lithuania | 308 | 37.7 | 32.5 | 19.5 | 10.4 | 3187 | 42101 | 41243 |
| Croatia | 315 | 34.6 | 37.1 | 15.9 | 12.4 | 4381 | 51444 | 50844 |
| Norway | 356 | 42.4 | 26.4 | 23.6 | 7.6 | 4804 | 41764 | 40886 |
| Finland | 355 | 42.8 | 35.2 | 16.6 | 5.4 | 5302 | 49869 | 48933 |
| Denmark | 420 | 40.0 | 37.4 | 13.6 | 9.0 | 5497 | 54446 | 53319 |
| Switzerland | 451 | 43.9 | 31.0 | 16.4 | 8.6 | 7558 | 62183 | 61384 |
| Austria | 374 | 35.8 | 25.4 | 23.5 | 15.2 | 8359 | 77256 | 75765 |
| Sweden | 457 | 43.3 | 30.6 | 19.5 | 6.6 | 9259 | 91504 | 90516 |
| Hungary | 482 | 36.1 | 27.6 | 29.3 | 7.1 | 10024 | 130830 | 129695 |
| Czech Republic | 443 | 31.8 | 35.9 | 23.0 | 9.3 | 10383 | 107448 | 106636 |
| Belgium | 517 | 37.3 | 33.7 | 20.9 | 8.1 | 10678 | 104344 | 103257 |
| Netherlands | 554 | 33.9 | 28.0 | 25.1 | 13.0 | 16489 | 138373 | 137645 |
| Romania | 444 | 31.8 | 36.0 | 19.4 | 12.8 | 21594 | 258000 | 257265 |
| Poland | 581 | 27.7 | 33.2 | 22.9 | 16.2 | 38114 | 375231 | 374295 |
| Spain | 672 | 33.0 | 33.9 | 18.2 | 14.9 | 44594 | 385512 | 383204 |
| UK | 710 | 30.8 | 33.8 | 21.1 | 14.2 | 61690 | 580733 | 578572 |
| France | 738 | 27.5 | 38.8 | 22.6 | 11.1 | 61709 | 535320 | 532293 |
| Germany | 791 | 27.8 | 35.4 | 24.7 | 12.1 | 81980 | 852566 | 849400 |

*Coded at ICD-10, three-position level; including all causes of death with at least five mean number of deaths in the 15-year period.
COD, cause of death; ICD, International Classification of Diseases.

estimated the index that represents the point closest to the top-left part of the ROC curve.[22 26]

All analyses were conducted using R statistical software V.3.5.1.[28]

**Patient and public involvement**

No patients were involved in this study.

**RESULTS**

The number of CODs with at least five annual deaths on average varied between 202 (Estonia) and 791 (Germany) (table 1). Of these CODs, 32.6%, 20.2% and 11.2% had a significant trend following a linear, quadratic or cubic model, respectively. The percentage of CODs with no significant trend (ie, constant model) varied from 27.5% to 43.9%, and was highest in the Nordic countries, Switzerland and Slovenia. More detailed information on the best model for each COD in each European country can be found in an additional file (online supplementary resource 1).

Both COD size and COD type were significantly associated with the likelihood of having a significant long-term trend (p<0.001) (table 2). For every 10% increase in the COD size, we observed a 18% increase (1.1ˆ1.73=0.18) in the odds of having a significant trend (OR=1.73, 95% CI=1.67 to 1.79). Regarding the COD type, neoplasms and digestive system diseases had lower probability for detecting a trend in comparison to the circulatory system diseases. On the other hand, this probability was higher for infectious diseases, mental diseases, and signs and symptoms. Figure 1 shows for each COD chapter in each country the estimated probability of having a significant long-term trend in relation to COD size. The variation between COD chapters was substantial, irrespective of

**Table 2**  Relationship between the likelihood for a COD to have a significant long-term trend with its size, corresponding ICD-10 chapter and country

| COD characteristic | | Number of CODs | Total number of deaths | OR* (95% CI) |
|---|---|---|---|---|
| **Size** | | | | |
| Log (mean deaths†) | | | – | **1.73** (1.67 to 1.79) |
| **ICD10 chapter** | | | | |
| I00.I99 | Diseases of the circulatory system | 988 | 23 610 116 | Reference |
| A00.B99 | Certain infectious and parasitic diseases | 454 | 823 839 | **1.63** (1.25 to 2.14) |
| C00.D48 | Neoplasms | 1871 | 15 632 543 | **0.57** (0.47 to 0.69) |
| D50.D89 | Diseases of blood and blood-forming organs and certain disorders involving the immune mechanisms | 217 | 143 156 | 0.82 (0.59 to 1.14) |
| E00.E90 | Endocrine, nutritional and metabolic diseases | 365 | 1 547 431 | 1.01 (0.76 to 1.34) |
| F00.F99 | Mental, behavioural disorders | 230 | 1 725 881 | **1.62** (1.13 to 2.30) |
| G00.G99 | Diseases of the nervous system | 536 | 1 888 629 | 0.90 (0.70 to 1.15) |
| J00.J99 | Diseases of the respiratory system | 553 | 4 740 481 | 1.22 (0.94 to 1.58) |
| K00.K93 | Diseases of the digestive system | 767 | 2 816 168 | **0.75** (0.59 to 0.94) |
| M00.M99 | Diseases of the musculoskeletal system and connective tissue | 367 | 283 108 | 1.14 (0.86 to 1.50) |
| N00.N99 | Diseases of the genitourinary system | 385 | 1 029 118 | 0.98 (0.74 to 1.29) |
| Q00.Q99 | Congenital malformations, deformations and chromosomal abnormalities | 325 | 141 848 | 0.89 (0.67 to 1.19) |
| R00.R99 | Symptoms, signs and abnormal clinical and laboratory findings, not elsewhere classified (signs-and-symptoms) | 258 | 2 062 587 | **1.68** (1.18 to 2.38) |
| V01.Y98 | External causes of morbidity and mortality | 1992 | 3 005 539 | 1.20 (0.99 to 1.46) |
| Other‡ | | 349 | 240 748 | 1.29 (0.97 to 1.71) |
| **Intra-class correlation for the country level** | | | | |
| model with fixed effects for size and ICD10 chapter | | | 0.003 | |
| model with no fixed effects | | | 0.013 | |

*ORs in bold were statistically significant with a p value <0.05.
†Mean deaths: mean of the annual number of deaths for a cause of death monitored in the 15-year period, measured per country. Only including CODs with five or more deaths.
‡'Other' consists of the causes of death classified in the ICD-10 chapters H00.H59: diseases of the eye and adnexa, H60.H95: diseases of the ear and mastoid process, L00.L99: diseases of the skin and subcutaneous tissue, O00.O99: Pregnancy, childbirth and the puerperium and P00.P96: certain conditions originating in the perinatal period.

COD size. Neoplasms (chapter C00.D48) as a group of CODs showed the lowest probability of having a detectable trend.

We found only small variation of countries in the likelihood of detecting a long-term trend, as the ICC for the country-level random effect was only 0.013 (without fixed effects for chapter and size) and 0.003 (with fixed effects) (table 2). Figure 2 illustrates the small differences between countries in the estimated probability of having a long-term trend.

Figure 3A describes the se and sp for detecting a significant long-term trend using different levels of thresholds in terms of any COD size. The AUC corresponding to these se and sp values was 0.706, with 95% CI: 0.695 to 0.716. The maximised sum index (Youden Index) was 32.7 annual deaths, with se 61.4% and specificity (sp) 70.3%. The minimum difference index was 27.5 annual deaths (se=65.5%, sp=65.5%). The closest top-left index was 29.4 annual deaths (se=64.0%, sp=67.5%) (figure 3A).

The corresponding analysis for the neoplasms yielded a similar ROC curve (AUC=0.703, 95% CI: 0.680 to 0.727) (figure 3B). The Youden Index was 70.4, with sensitivity 61.5% and specificity 69.6%, and the minimum difference index was 60.5 annual deaths (se=64.8%, sp=64.8%). The closest top-left index was also 60.5 annual deaths (se=64.1%, sp=66.9%). Infectious and parasitic diseases (AUC=0.706, 95% CI: 0.695 to 0.716) yielded a Youden Index of 19.7 annual deaths, with sensitivity 67.1% and specificity 69.8%. The closest top-left index was identical, while the minimum difference

  Mitratza M, *et al. BMJ Open* 2020;**10**:e031702. doi:10.1136/bmjopen-2019-031702

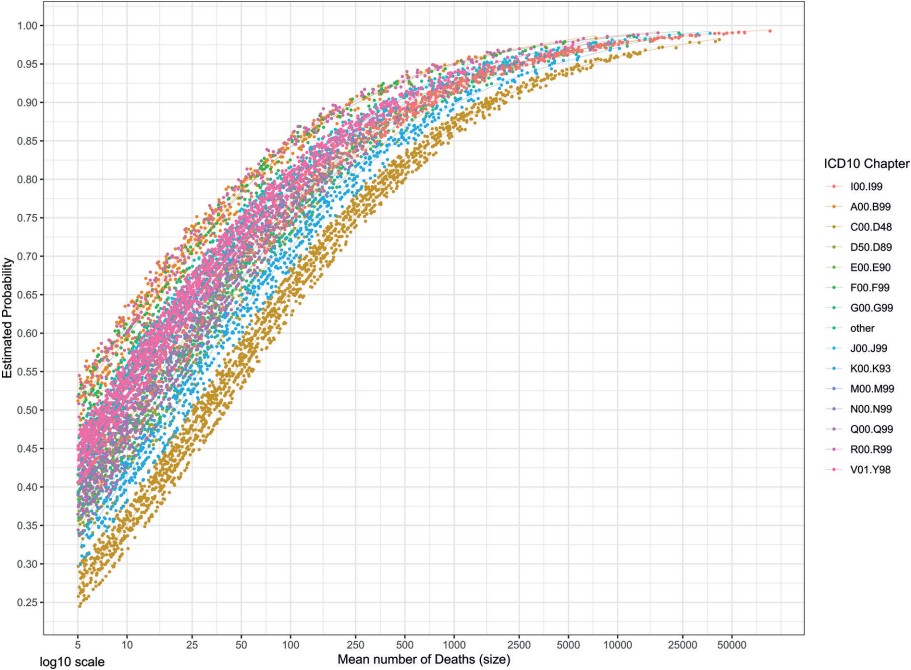

**Figure 1** Estimated probability for an underlying cause of death to have a significant long-term trend according to its size, by ICD-10 chapter. See table 2 for the definition of the chapters. ICD, International Classification of Diseases.

threshold was 19.3 annual deaths (se=67.4%, sp=67.4%) (figure 3C).

## DISCUSSION

COD data are used in widely varying settings, ranging from detailed mortality profiles to macro estimates. Applications include studies in localised areas,[29] single countries[30] or worldwide[2 31]; for a single-disease[32 33] or disease group[9]; monitored for days or a long-term period[7]; for specific age groups[33 34] or specific situations (eg, maternal mortality,[5] external causes[35–37]). These settings all impose different requirements on the collected data. Here we focused on one particular application: national estimates of mortality time trends for a reasonably long period (15 years), for a considerable number of countries (21) that have quite comparable CODs data collection and

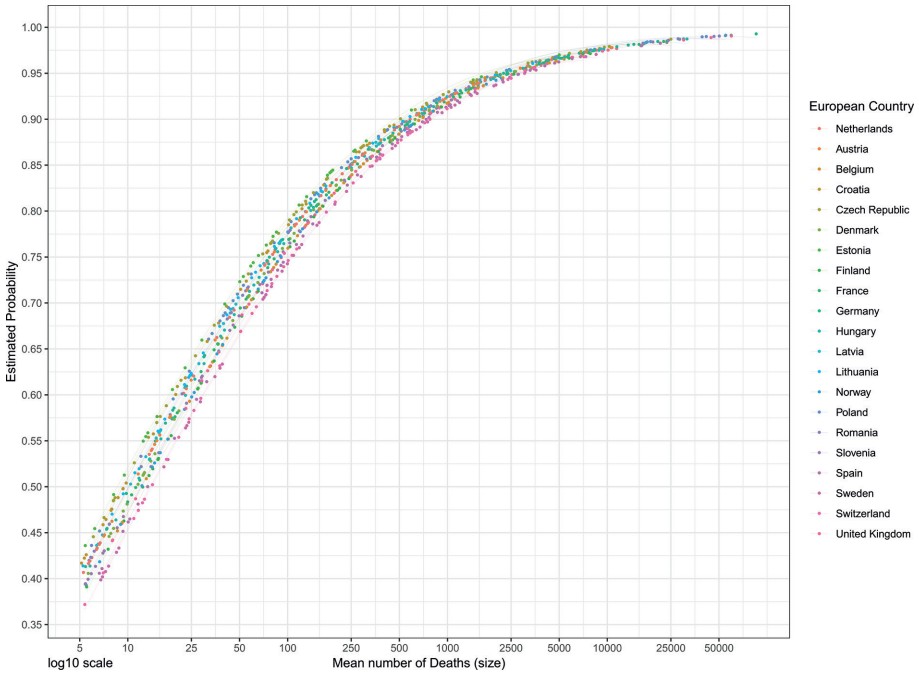

**Figure 2** Estimated probability for a disease of the circulatory system to have a significant long-term mortality trend according to its size, by European country.

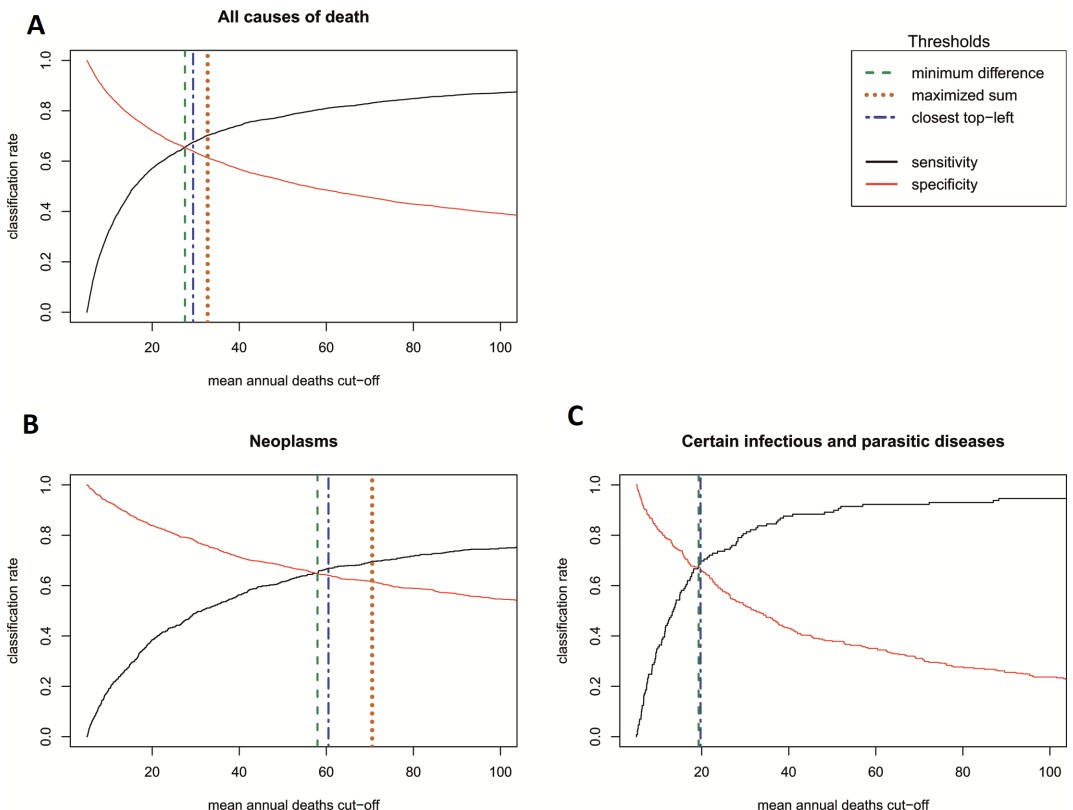

**Figure 3** Sensitivity and specificity of the cause of death size for the detection of significant long-term time-trends, with thresholds for the optimal cause of death size for trend analysis.

registration systems,[38] covering as many CODs as possible. Our aim was to investigate the effect of the size of a COD on the probability to detect a significant trend, and how this is related to country and type of COD (ICD-10 chapter).

Our results indicate that both the size and the type of a COD were associated with the probability of detecting a significant trend, while variations among European countries were negligible. Some types of CODs, particularly neoplasms and digestive system diseases, had a lower probability for detecting a significant trend in comparison to the circulatory system diseases, whereas infectious and mental diseases had a higher probability. The results suggest a general size criterion of 30 annual deaths for selecting CODs to include in long-term mortality trends analysis, and a more specific criterion of 65 deaths for neoplasms and 20 for infectious diseases.

We should outline the limitations of our study. First, due to the exclusion of CODs with less than five annual deaths on average, smaller countries were represented in our analysis with fewer CODs. However, this is unlikely to have a strong influence on the results, as the suggested COD size threshold of about 30 deaths is much higher than the lower limit of 5 mean annual deaths. Second, although we proposed the COD size as a criterion to select CODs for long-term trend analysis, we acknowledge that other criteria could be used, such as greater preference to CODs that involve high healthcare costs or that are potentially modifiable by preventive or curative actions. Third,

the likelihood to demonstrate a time trend with statistical significance depends on the statistical method that is used to describe these trends. Our results are dependent on the balance between avoiding type I error and type II errors. As for type I errors, we chose a significance level of $\alpha=0.05$. A more restrictive significance level would have the consequence to increase type II errors, that is, to reduce the proportion of CODs for which a trend would be detected based on our method.

Moreover, our results should be seen as conditional on our use of OLS models with polynomial terms. The OLS approach may not be appropriate for small counts. However, the approximation of a Poisson by a normal error distribution is generally assumed to be adequate if the mean number of observations is about five or more. For larger counts, OLS has the benefit that a variance can be estimated, rather than postulated.

In addition, an alternative to the classic polynomial regression approach would have been to use Generalized Additive Models (GAMs). These models have the advantage of being able to pick up trends that are not polynomial. In a sensitivity analysis, we applied GAMs with Gaussian process smoothing function to our data. We found that a long-term trend could be detected in 71.7% of the CODs, as compared to 64.0% in our original analysis. There were virtually no CODs for which a trend could be detected when using polynomial models but not when using GAMs. This would imply that our results are approximately robust to the method used, although somewhat conservative.

Finally, including spatial correlation in our model may have altered the chance of detecting a significant trend for CODs with marked geographical patterns. We calculated Moran's I test for spatial correlation among countries regarding the proportion of CODs in each country with a detected long-term trend. The Moran's I test was found to be not statistically significant for all CODs collectively (p value=0.988). At the level of COD chapters, we found significant spatial correlation for the chapters C–D (p value=0.002), E (p value=0.025), and V–Y (p value=0.001), but not for other chapters.

We found that mortality from neoplasms was less likely to have a significant trend, for a given size of COD. This may relate to the fact that the neoplasm mortality levels tend to change gradually over time, without short-term trend changes.[6] Additionally, cancers are usually coded reliably and consistently over time,[39–41] so that coding artefacts can rarely induce artificial changes. Conversely, the dynamic nature of infectious diseases may be responsible for their higher likelihood to change over time, and to have significant trends even with relatively small numbers of deaths. Similarly, the chapter of signs-and-symptoms is sensitive to changes in the coding rules and practices, thus creating significant changes even with small number of deaths.

Our study showed that European countries did not vary substantially in the probability of detecting a significant long-term trend in CODs of the same size and type. This finding is surprising given the heterogeneity of the countries in terms of demographic characteristics, disease epidemiology, healthcare systems and coding practices. We found that differences between countries in the proportions of CODs with a significant trend (table 1) can be related to differences in COD size which is strongly related to the differences in population size. Consequently, our analysis provides support for establishing one common COD size threshold, applicable for all European countries and for use in international trend analyses.

Currently, there is no gold standard for the selection of CODs to analyse for long-term trends. In this study, we attempted to set such a standard, based on the criterion of the COD size which is easy to measure for each single COD. We calculated thresholds with three common methods which came close enough (eg, in the range of 28 to 33 deaths) to support one general recommendation for practical use. Of course, different thresholds may be preferred, depending on the user's preference to avoid either false positives (by selecting a higher threshold) or false negatives (lower threshold).

In our data, the number of CODs that surpassed our recommended threshold of 30 annual deaths on average was around 500 for the biggest countries, 200–250 for the middle-sized countries and around 100 for the smaller European countries (results not shown). In total, 52 CODs had over 30 annual deaths on average in each country included in our analysis. This implies that at least 52 CODs could be included in the international comparison of long-term trends, but up to 100 if one is to accept a greater risk of false positives in smaller countries.

From the public health practitioner's perspective, the findings of our study can be used in order to set realistic expectations about the number of CODs that are likely to have a significant long-term trend in populations. We recommend a size criterion of 30 annual deaths to be considered when planning for national or international monitoring and comparisons of cause-specific mortality.

**Contributors** AEK, JWPFK and MM designed the study. MM performed the calculations. MM, AEK and JWPFK analysed and interpreted the results and formulated the conclusions, and made major contributions to the manuscript. All authors read and approved the final manuscript.

**Funding** The authors have not declared a specific grant for this research from any funding agency in the public, commercial or not-for-profit sectors.

**Competing interests** None declared.

**Patient consent for publication** Not required.

**Ethics approval** No ethical approval was required for this study, as no living subjects were involved and only aggregated administrative data were used.

**Provenance and peer review** Not commissioned; externally peer reviewed.

**Data availability statement** Data are available in a public, open access repository.

**ORCID iD**
Marianna Mitratza http://orcid.org/0000-0002-1573-2015

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
