## [Reviewer comments · BMJ Open]

ARTICLE DETAILS

TITLE (PROVISIONAL)	How large should a cause of death be in order to be included in mortality trend analysis? deriving a cut-off point from retrospective trend analyses in 21 European countries
AUTHORS	Mitratza, Marianna; Kardaun, Jan W.P.F.; Kunst, Anton E

VERSION 1 – REVIEW

REVIEWER	Kate Brameld Curtin University Perth Australia
REVIEW RETURNED	06-Jul-2019

GENERAL COMMENTS	Regarding "What is known on this subject" I don't think the second sentence accurately describes the current situation. Maybe need something like " this approach does not take account of the statistical likelihood of observing a trend". Page 8, line 3 of second paragraph (line numbers are cut off in my copy) needs rewording. Currently states "the and the binary" Page 11, Lline 7 before Fig 1, Sentence finishes with "had higher" - this doesn't make sense. Page 15 , second sentence of fist paragraph " mortality from neoplasms tend to develop rather gradually throughout the years". I don't understand what this means. On reviewing the reference number 6, this states that cancer mortality trends differ between high and low income countries, which suggests that the analysis should include this variable (high vs low income countries), as the differing trends may be cancelling each other out in the analysis presented in this paper. Could the authors look at this possibility?
---

REVIEWER	Pavel Chernyavskiy Department of Mathematics and Statistics, University of Wyoming, USA
REVIEW RETURNED	28-Sep-2019

GENERAL COMMENTS	Thank you for the opportunity to review this manuscript. The authors undertake an ambitious effort to determine an event count at which patterns in mortality could be detected. The ultimate goal is to help set realistic expectations – a goal that I think is both important and necessary to quantify. However, I am concerned by some inexact language at several points in the manuscript that sounds like the authors are performing a type of power analysis, when that isn't really what they are doing. In addition, I think the
--

article can benefit from moving away from using polynomials to model trends over time towards more modern and likely more robust Generalized Additive Model-based approach. Finally, the authors analyze data across (partially-)adjacent countries without considering or testing for the presence of spatial correlation. It seems very likely that this correlation will materialize in at least the mean rates, if not also the trends over time that the authors are more interested in.

Major Comments

- pg. 5 (lines 14-48). I am not sure I follow the purpose of this paragraph. One of the main factors that influences the likelihood of concluding a trend exists - if it is in fact present - is the exact statistical method used. For example, when analyzing mortality counts, a log-transformed OLS or WLS model will have different power vs. a Poisson or Negative-Binomial GLM. In addition, if looking at a collection of European (for instance) countries, one must consider including spatial correlation among mean rates and trends, which may help reduce coefficient uncertainty and help detect a trend. Finally, for rare events with many 0s, there are several zero-inflated or hurdle regression methods, each with a different potential to detect a trend. Are the authors focusing on a specific regression model? I'm not yet clear on when this would be helpful because investigators pick their statistical based on their exact dataset in question, based in part, for example based, on how many 0s are present and whether there is over-dispersion in the data.

- pg. 5 (lines 51-52). I read the authors' statement "likelihood to detect a long-term trend" as closely related to the notion of statistical power. I would recommend that the authors use more precise language in their statements reg. power, because (generally) we only want to detect a trend if it is in fact present. Detecting a trend that is not present is of course an example of Type I error. As for an empirical assessment, it is not uncommon for investigators to undertake a power analysis either prior to data collection or post-data collection. However, as I have already stated, any power analysis is closely tied to the exact statistical method used and the desire to balance Type I error-rate alongside Type II error-rate and power.

- Pg. 6 (lines 9 – 20). I think I understand what the authors are trying to achieve, but I'm not sure this is possible without narrowing down their statistical methodology to at least a class of statistical models. For example, the authors could use Poisson, Negative Binomial, Zeroinflated Poisson and the Zero-inflated Negative Binomial (implementable using widely available R packages) to determine the event threshold they seek. Without this, or a stated desired level of power and type I error rate, I'm not sure it is possible to do what the authors are seeking.

- Pg. 7 (lines 9 – 41). It seems the authors exclude rare causes of deaths that have a lot of 0 counts. I'm not sure this appropriate if the stated goal of the paper is to find a minimum cutoff of event counts to detect a trend. In addition, I would urge the authors to be more precise in Referee Report bmjopen-2019-031702

specify the exact regression models they used (line 18). I am assuming this was a either Poisson or Negative Binomial regression? It does seem, however, that by eliminating COD with many 0s the authors do narrow down their methodology to exclude zero-inflated and hurdle models. I think this is OK, as long as this is clearly stated somewhere.

	- Pg. 7 (lines 24-25). Polynomial regressions often impose unrealistic shapes on time trends and can be strongly affected by sparse data. I understand what the authors are trying to determine using the AICc, but it seems Generalized Additive Models (GAMs) may be better suited for this. GAMs could, for example, provide an approximate test of the non-linear coefficient terms and have the ability to pick up patterns that are not polynomial. Finally, at least in my view, GAMs are becoming more prevalent across statistical analyses and so I would expect these to compete with a classic polynomial regression approach. Have the authors considered this as an alternative? Perhaps, the authors could include it as a sensitivity analysis to test the robustness of their findings? - Pg. 7 (line 45). I understand and support using multi-level models to analyze data collected across countries, but have the authors examined the presence of spatial correlation among adjacent countries? It would seem very likely spatial correlation would be present at least in the mean rates, if not both mean rates and coefficients that control the shape of trends over time. Note that this would likely alter model AICc, and possibly the polynomial regression order chosen. It would be important to assess this spatial correlation in estimating the trend models as well as the multi-level logistic regression. Minor Comments - pg. 3. I would prefer that the authors used more precise language in the Abstract. For example, what is the meaning of “potentially useful” (line 9)? Are we thinking about some desired threshold of signal-to-noise? Are thinking about some limit of missing data? On line 33 it would be appropriate to specify the type of regression analysis performed as there are many types. It seems the authors fit polynomial regressions, correct? On line 35 the authors use the word “demonstrable long-term trend”. What is the meaning of “demonstrable”? Is it statistical significance at some alpha level? - Pg.4 line 3: It seems the authors are using “demonstrable” and “potential to detect” to mean the same thing. With “potential to detect”, at least statistically, the authors seem to be referring to power or Type II error. Again, I would prefer more precise language about what they mean. If they indeed mean power, then what level of power is desired (80%?). - Pg. 27 (line 5). Was this article submitted to Journal of Epidemiology and Community Health or BMJ Open? - pgs. 29-804 of the PDF file are strangely-formatted. This was likely an error of some sort that occurred during type-setting. I just wanted to bring this to the authors’ attention.
--	---

VERSION 1 – AUTHOR RESPONSE

- Response to Reviewer #1:**
 Reviewer Name: Kate Brameld
 Institution and Country: Curtin University Perth Australia

Comment 1:

Regarding "What is known on this subject" I don't think the second sentence accurately describes the current situation. Maybe need something like " this approach does not take account of the statistical likelihood of observing a trend".

Reply 1: We have already removed the section "What is known on this subject", after requested by the editor due to not being required for the journal.

Comment 2:

Page 8, line 3 of second paragraph (line numbers are cut off in my copy) needs rewording. Currently states "the and the binary"

Reply 2: We thank the reviewer for alerting us to this error. In line 186 of the new manuscript (version with track changes) we have made this correction.

Comment 3:

Page 11, Line 7 before Fig 1, Sentence finishes with "had higher" - this doesn't make sense.

Reply 3: We would like to thank the reviewer for alerting us to this error. In line 225 of the new manuscript (version with track changes) we have made this correction.

Comment 4:

Page 15, second sentence of first paragraph "mortality from neoplasms tend to develop rather gradually throughout the years". I don't understand what this means. On reviewing the reference number 6, this states that cancer mortality trends differ between high and low income countries, which suggests that the analysis should include this variable (high vs low income countries), as the differing trends may be cancelling each other out in the analysis presented in this paper. Could the authors look at this possibility?

Reply 4: We would like to clarify that by this sentence we mean that we can see a generally gradual rather than sharply fluctuating development of cancer mortality. In reference 6, particularly in the Supplement showing the mortality trends from different cancer types, this gradual pattern is common in both high and low income countries, although we recognize it may be increasing, decreasing, or mixed for different countries. We clarified this by changing the sentence into "This may relate to the fact that the neoplasm mortality levels tend to change gradually over time, without short-term trend changes", in line 315 of the new manuscript (version with track changes).

- **Response to Reviewer #2:**

Reviewer Name: Pavel Chernyavskiy

Institution and Country: Department of Mathematics and Statistics, University of Wyoming, USA

Major Comments

Comment 1:

pg. 5 (lines 14-48). I am not sure I follow the purpose of this paragraph.

- a) One of the main factors that influences the likelihood of concluding a trend exists - if it is in fact present - is the exact statistical method used. For example, when analyzing mortality counts, a log-transformed OLS or WLS model will have different power vs. a Poisson or Negative-Binomial GLM.*
- b) In addition, if looking at a collection of European (for instance) countries, one must consider including spatial correlation among mean rates and trends, which may help reduce coefficient uncertainty and help detect a trend.*
- c) Finally, for rare events with many 0s, there are several zero-inflated or hurdle regression methods, each with a different potential to detect a trend. Are the authors focusing on a specific regression model? I'm not yet clear on when this would be helpful because investigators pick*

their statistical based on their exact dataset in question, based in part, for example based, on how many 0s are present and whether there is over-dispersion in the data.

Reply 1: We thank the reviewer for these comments, which we ordered into three comments (a, b, c). We would like to respond to each comment separately.

- a) We agree that the possibility of detecting a trend depend on the statistical method used. In the revised manuscript, we make explicit that our results are conditional on the method that we used in line 295 of the new manuscript (version with track changes).
- b) For the issue of spatial autocorrelation, we would like to refer to our response to comment 6 below.
- c) For this issue, we would like to refer to our response to comment 4 below.

Comment 2:

pg. 5 (lines 51-52). I read the authors' statement "likelihood to detect a long-term trend" as closely related to the notion of statistical power. I would recommend that the authors use more precise language in their statements reg. power, because (generally) we only want to detect a trend if it is in fact present. Detecting a trend that is not present is of course an example of Type I error. As for an empirical assessment, it is not uncommon for investigators to undertake a power analysis either prior to data collection or post-data collection. However, as I have already stated, any power analysis is closely tied to the exact statistical method used and the desire to balance Type I error-rate alongside Type II error-rate and power.

Reply 2: We agree that our analysis has similarities with power analysis, but we would like to clarify that we are not performing a power analysis in a formal way, but in a descriptive, empirical way, in which the outcome also depends on the magnitude of the real trend in country- and cause-specific mortality. We also agree that the outcomes of our analysis, like a formal power analysis, are tied to the method used (see our reply above to comment 1) and to the balance between Type I and Type II error. We added this latter point to our discussion in lines 297-301 of the new manuscript (version with track changes).

Comment 3:

Pg. 6 (lines 9 – 20). I think I understand what the authors are trying to achieve, but I'm not sure this is possible without narrowing down their statistical methodology to at least a class of statistical models. For example, the authors could use Poisson, Negative Binomial, Zero-inflated Poisson and the Zero-inflated Negative Binomial (implementable using widely available R packages) to determine the event threshold they seek. Without this, or a stated desired level of power and type I error rate, I'm not sure it is possible to do what the authors are seeking.

Reply 3: We would like to refer to comments 2, 4 and 5 for our replies, as we believe they address the same issue of specifying the methods which are used.

Comment 4:

Pg. 7 (lines 9 – 41). It seems the authors exclude rare causes of deaths that have a lot of 0 counts. I'm not sure this appropriate if the stated goal of the paper is to find a minimum cutoff of event counts to detect a trend. In addition, I would urge the authors to be more precise in specify the exact regression models they used (line 18). I am assuming this was a either Poisson or Negative Binomial regression? It does seem, however, that by eliminating COD with many 0s the authors do narrow down their methodology to exclude zero-inflated and hurdle models. I think this is OK, as long as this is clearly stated somewhere.

Reply 4: We would like to confirm that we had excluded rare causes of death, with less than 5 deaths per year on average. We think that this is appropriate, given that we expect that a time-trend cannot be detected for almost any of such rare causes (assuming a significance level of 0.05 for type I errors). We recognise that the methods section may not describe the models used clearly enough. We have now clarified that we used ordinary least squares regression models, in line 157 of the new manuscript (version with track changes). To this, we would like to add that in sensitivity analysis that we made for Dutch data, almost the same results were obtained when using Poisson regression models (we added the new reference [18]).

Comment 5:

Pg. 7 (lines 24-25). Polynomial regressions often impose unrealistic shapes on time trends and can be strongly affected by sparse data. I understand what the authors are trying to determine using the AICc, but it seems Generalized Additive Models (GAMs) may be better suited for this. GAMs could, for example, provide an approximate test of the non-linear coefficient terms and have the ability to pick up patterns that are not polynomial. Finally, at least in my view, GAMs are becoming more prevalent across statistical analyses and so I would expect these to compete with a classic polynomial regression approach. Have the authors considered this as an alternative? Perhaps, the authors could include it as a sensitivity analysis to test the robustness of their findings?

Reply 5: We would like to thank the reviewer for this useful suggestion to consider the use of GAMs. We have conducted a sensitivity analysis using these models, and we report its results in the Discussion section, lines 303-312 of the new manuscript (version with track changes). In general, we found that GAMs are able to detect trends in a greater number of COD. This may reflect that GAMs can accommodate more smoothing functions. On the other hand, with only a limited number of data points, as in our study, there may be a risk of overfitting with GAMs. Taken all together, we think that our general conclusion may be approximately robust to the method used, or be on the conservative side.

Comment 6:

Pg. 7 (line 45). I understand and support using multi-level models to analyze data collected across countries, but have the authors examined the presence of spatial correlation among adjacent countries? It would seem very likely spatial correlation would be present at least in the mean rates, if not both mean rates and coefficients that control the shape of trends over time. Note that this would likely alter model AICc, and possibly the polynomial regression order chosen. It would be important to assess this spatial correlation in estimating the trend models as well as the multi-level logistic regression.

Reply 6:

To our understanding, spatial autocorrelation is usually taken into account in ecological analysis that aims to assess an association between two or more variables across geographical areas, and that should adjust for interdependency between areas in order to adjust variance estimates. In our case, however, we are primarily interested in trends over time, instead of variations between countries. Most of our analysis are done for individual countries. Therefore, we thought that spatial autocorrelation of no or little relevance to the topic of our paper.

Yet, we have aimed to test for spatial autocorrelation in our data. When assessing spatial autocorrelation, there are multiple ways to take into account the complex geography of countries within Europe. Moreover, there are multiple ways in which we could measure the dependent variables (mortality levels, whether trends can be detected, magnitude of change). As a simple test, we calculated Moran's I test for spatial correlation among these countries regarding the proportion of COD in each country with a detected long-term trend. The Moran's I test was found to be not statistically significant for all CODs collectively (p-value = 0.988). At the level of COD chapters, we found significant spatial

correlation for the chapters C00.D48 (p-value = 0.002), E00.E90 (p-value = 0.025), and V01.Y98 (p-value = 0.001), but not for other chapters.

All in all, given the considerations and findings above, we think that there is no reason to address the issue of spatial autocorrelation in the paper.

Minor Comments

Comment 7:

pg. 3. I would prefer that the authors used more precise language in the Abstract. For example, what is the meaning of “potentially useful” (line 9)? Are we thinking about some desired threshold of signal-to-noise? Are thinking about some limit of missing data? On line 33 it would be appropriate to specify the type of regression analysis performed as there are many types. It seems the authors fit polynomial regressions, correct? On line 35 the authors use the word “demonstrable long-term trend”. What is the meaning of “demonstrable”? Is it statistical significance at some alpha level?

Reply 7: We thank the reviewer for these comments. We have made corrections in lines 53,54,58,59,61 of the new manuscript (version with track changes), as well as similar encountered occasions throughout the manuscript.

Comment 8:

Pg.4 line 3: It seems the authors are using “demonstrable” and “potential to detect” to mean the same thing. With “potential to detect”, at least statistically, the authors seem to be referring to power or Type II error. Again, I would prefer more precise language about what they mean. If they indeed mean power, then what level of power is desired (80%?).

Reply 8: We thank the reviewer for this suggestion. We have rephrased this part in lines 67-68 of the new manuscript (version with track changes). We would like to clarify that we did not perform a formal power analysis, but an empirical descriptive analysis in which we could not assess an a priori power level.

Comment 9:

Pg. 27 (line 5). Was this article submitted to Journal of Epidemiology and Community Health or BMJ Open?

Reply 9: We thank the reviewer for this notification. This must be an error during the automatic transfer of the manuscript from the Journal of Epidemiology and Community Health to BMJ Open in the system. We have corrected this.

Comment 10:

pgs. 29-804 of the PDF file are strangely-formatted.

Reply 10: We agree with the reviewer that the PDF file has spread the supplementary material in too many pages, making it hard for the readers to get a proper overview. To avoid this, we have re-formatted the PDF file.

VERSION 2 – REVIEW

REVIEWER	Pavel Chernyavskiy University of Wyoming, USA
REVIEW RETURNED	08-Dec-2019

GENERAL COMMENTS	Thank you for the opportunity to review a revision of this manuscript. I also thank the authors for taking the time to carefully and thoroughly address most of my comments. However, because their findings are conditional on the statistical method used, and the authors have now described this method in more detail, I still have a few concerns I would like to bring up. I believe that these comments can be addressed by adding some text to Methods and/or Discussion, so I do not require to review another set of revisions. Thank you for the opportunity to review a revision of this manuscript. I also thank the authors for taking the time to carefully and thoroughly address most of my comments. However, because their findings are conditional on the statistical method used, and the authors have now described this method in more detail, I still have a few concerns I would like to bring up. I believe that these comments can be addressed by adding some text to Methods and/or Discussion, so I do not require to review another set of revisions. Additional Comments  - Reply to Comment 4 and pg. 6 (lines 146-150). It appears the authors use ordinary least squares (OLS) and performed a sensitivity analysis with a Poisson on one country. How did the authors account for different population at risk for different countries and years? Sometimes, one can include this as a weight in OLS models and produce reasonable results. I also see that the authors have cited a paper where they compared the OLS approach to the Poisson approach and produced similar results. This can occur if the counts large and so the mean count is far enough away from 0 to avoid issues. My concern is that the OLS model is not generally appropriate for counts and I am afraid that if the analysis section remains as is, other readers who cite the current manuscript will apply OLS regression to counts, possibly incorrectly. I have no reason to doubt the authors' sensitivity analysis, but it would be nice to include text in the current paper to warn others that the OLS approach may fail when applied to counts, especially small counts. - Reply to Comment 6. I am in total agreement that there are many ways of specifying the neighborhood structure of Europe. However, by omitting a potentially important spatial correlation, predictor standard errors – and therefore the likelihood of detecting a significant trend – can be affected. I would recommend including text in the discussion warning readers that including spatial correlation in their model can alter the chances of detecting a significant trend.
--

VERSION 2 – AUTHOR RESPONSE

• Response to Reviewer:

Reviewer Name: Pavel Chernyavskiy

Institution and Country: Department of Mathematics and Statistics, University of Wyoming, USA

Additional Comments

Comment 1 - Reply to Comment 4 and pg. 6 (lines 146-150) :

It appears the authors use ordinary least squares (OLS) and performed a sensitivity analysis with a Poisson on one country. How did the authors account for different population at risk for different

countries and years? Sometimes, one can include this as a weight in OLS models and produce reasonable results. I also see that the authors have cited a paper where they compared the OLS approach to the Poisson approach and produced similar results. This can occur if the counts large and so the mean count is far enough away from 0 to avoid issues. My concern is that the OLS model is not generally appropriate for counts and I am afraid that if the analysis section remains as is, other readers who cite the current manuscript will apply OLS regression to counts, possibly incorrectly. I have no reason to doubt the authors' sensitivity analysis, but it would be nice to include text in the current paper to warn others that the OLS approach may fail when applied to counts, especially small counts.

Reply 1: We thank the reviewer for this comment. We have addressed the change in age composition in each country over the 15-year period, by standardizing the mortality counts to the age composition of the European standard population 2013 and weighting to the middle period population of each country, as stated in the Methods section. Moreover, we have not weighted for differences in population at risk between countries because, given the research questions, we considered each country as an equally important unit of observation, irrespective of the population size of the country. In addition, in lines 292-298 of the new manuscript (version with track changes), we have added text explaining clearly that the OLS approach may not be appropriate for small counts.

Comment 2 - Reply to Comment 6 :

I am in total agreement that there are many ways of specifying the neighborhood structure of Europe. However, by omitting a potentially important spatial correlation, predictor standard errors – and therefore the likelihood of detecting a significant trend – can be affected. I would recommend including text in the discussion warning readers that including spatial correlation in their model can alter the chances of detecting a significant trend.

Reply 2: We agree that the aspect of spatial correlation could be explained more in our text. In lines 309-315 of the new manuscript (version with track changes), we have added a new paragraph in the Discussion section. We state that including parameters of spatial correlation in our models may have altered the chance of detecting a significant trend. However, we believe that this influence would affect only specific CODs.